

# Bioactivity of compounds secreted by symbiont bacteria of Nudibranchs from Indonesia

Rhesi Kristiana[1], Gilles Bedoux[2], Gerard Pals[3], I. Wayan Mudianta[4], Laure Taupin[2], Christel Marty[2], Meezan Ardhanu Asagabaldan[1], Diah Ayuningrum[1,8], Agus Trianto[5], Nathalie Bourgougnon[2], Ocky Karna Radjasa[5], Agus Sabdono[5] and Muhammad Hanafi[6,7]

[1] Department of Coastal Resources Management, Universities Diponegoro, Semarang, Central Java, Indonesia
[2] Laboratory of Marine Biotechnology and Chemistry, Université de Bretagne Sud, Vannes, Bretagne, France
[3] Center for Connective Tissue research, VU University medical center, Amsterdam, The Netherlands
[4] Chemical Analysis Study Program, Universitas Pendidikan Ganesha, Singaraja, Bali, Indonesia
[5] Department of Marine Sciences, Faculty of Fisheries and Marine Sciences, Universities Diponegoro, Semarang, Central Java, Indonesia
[6] Research Center for Chemistry, Indonesian Institute of Sciences., Tangerang Selatan, Banten, Indonesia
[7] Faculty of Pharmacy, Pancasila University, Srengseng Sawah Jakarta Selatan, Indonesia
[8] Department of Aquatic Resource Management, Diponegoro University, Semarang, Central Java, Indonesia

Corresponding author
Rhesi Kristiana,
rhesiundip@student.undip.ac.id

## ABSTRACT

The aims of this work are to isolate bacterial symbionts from nudibranchs and subsequently to determine anti-Methicillin resistant *Staphylococcus aureus* (MRSA), cytotoxicity and anti-Herpes simplex virus type 1 (HSV-1) activities of bio compounds. A total of 15 species of nudibranchs were collected from Karimunjawa and five species from Bali, respectively. A total of 245 bacteria isolates were obtained. The anti-MRSA activity screening activity indicated two active bacteria. Ethyl acetate extracts from supernatants, indicating extracelullar compounds, showed an inhibition zone against MRSA at concentrations of 500–1,000 μg/ml. DNA sequence analysis showed that the strain KJB-07 from *Phyllidia coelestis* was closely related to *Pseudoalteromonas rubra*, whereas the strain NP31-01 isolated from *Phyllidia varicosa* was closely related to *Virgibacillus salarius*. The extract of *Pseudoalteromonas rubra* was cytotoxic to Vero cells at a concentration of 75 μg/ml. The extract of *V. salarius* presented no cytotoxicity at concentrations of 5–1,000 μg/ml. No anti HSV-1 was observed for both isolated bacteria. This is the first study describing research on anti-MRSA, cytotoxicity and anti HSV-1 activity of bacterial symbionts from the viscera of nudibranch. Compounds produced by *Pseudoalteromonas rubra* and *V. salarius*, had potential anti-MRSA activity. However, only extracts from *Pseudoalteromonas rubra* showed cytotoxic effects on Vero cells. Three compounds were identified by LC/MS after purification from culture supernatant.

## INTRODUCTION

The emergence or resurgence of numerous infectious diseases has strongly risen up year by year between 1990 and 2013 (*Chen et al., 2019*). Recent reports describe that infectious diseases are currently becoming an alarming issue as they can spread from person to person, or from animal to a people (*Christiansen, 2018*). Infectious diseases are caused by four main kinds of germs, which are bacteria, viruses, fungi and parasites. Bacteria are the primary cause of diseases. During the past four decades, bacterial resistance to antibiotics has become a growing problem. For example, one type of bacteria, has evolved from a controllable nuisance into a serious public health concern. This bacterium is known as Methicillin-resistant *Staphylococcus aureus* (MRSA). Approximately one-third of the world population has *S. aureus* bacteria on their bodies at any given time. How the presence of *S. aureus* become a major human pathogen is still an unresolved issues (*Rasigade & Vandenesch, 2013*) since the bacteria can be present without causing an active infection. According to the Centers for Disease Control and Prevention, approximately 1% of all people have MRSA. There are two ways in which the infection can be acquired: hospital-acquired MRSA or community-associated MRSA. MRSA infections have caused higher morbidity and mortality compared to non-resistant strains (*Akhi et al., 2017*). The rapid emergence of MRSA infection is becoming critical, as the effectiveness of treatment of MRSA infection decreases due to the inability of bacteria to respond to antibiotics (*Guzmán-Blanco et al., 2009*). According to the World Health Organization, resistance to first line drugs to treat infections caused by *S. aureus* is widespread. Moreover 64% of infected people died due to MRSA infection (*WHO, 2017*).

The second cause of disease is viral infection. Herpes simplex virus (HSV), a DNA virus, is a common human pathogen with between 60–95% of certain populations infected with Herpes simplex virus type 1 (HSV-1), and between 6–50% infected with Herpes simplex virus type 2 (HSV-2). Primary and recurrent herpes virus infections in humans represent major risk factors for acquisition of primary HIV-1 infection (*Ben Sassi et al., 2008*). Acyclovir is the antiviral treatment of choice, but resistance to acyclovir has been reported due to *UL23* gene mutations (*Mitterreiter et al., 2016*). Hence, there is a need to develop new natural sources of agents for the management of HSV infections.

The Indonesian archipelago is a hotspot for biodiversity in the world (*Gastropoda et al., 2018*). Studies regarding the screening of secondary metabolites-producing bacterial symbionts are important for understanding their biotechnological potential (*Radjasa et al., 2011*). A few reports have been documented on the potential of bacterial symbionts from Indonesian marine invertebrates, such as coral (*Radjasa et al., 2007a*), soft coral (*Seyed Vahid Shetab-Boushehri, 2012*) and sponge (*Radjasa et al., 2007b*). Other studies using molluscs, for example, have reported that Serpin serine protease produced by *Octopus ocellatus* has potential antibacterial activity (*Wei et al., 2015*), that two gastropods

*Cerastoderma edule* and *Ostrea eduli*s present potential antibacterial and antiviral molecules (*Defer, Bourgougnon & Fleury, 2009*), and that tambjamine D an alkaloid isolated from the *Tambja eliora*, may have cytotoxic and genotoxic activity (*De Oliveira et al., 2007*). Furthermore the bioactive alkaloid ergopeptine has been extracted from *Pleurobranchus forskalii* (*Wakimoto, Tan & Abe, 2013*). However, secondary metabolites have only been investigated from a small proportion (<1%) of all Mollusca species (*Benkendorff, 2010*).

Nudibranchs, a group of marine gastropods, are of great interest for research and development of bioactive natural metabolites (*Avila et al., 2000*). These shell-less invertebrates are exposed to predators and to surface colonization by microorganisms. For them, the only chance of surviving and fighting effectively against these enemies is to produce compounds that are toxic for predators and infectious bacteria (*Barsby, Linington & Andersen, 2002*). Since marine tropical diversity also reflects chemical diversity, isolation of under-exploited bacterial symbionts from Indonesian nudibranchs offers a great opportunity for discovering novel bio-compounds, based on screening against various disease targets. Natural products from cultured symbionts of marine invertebrates present an opportunity to be developed in ways that circumvent the environmental and supply problems of pharmaceutical substances.

Several drugs from molluscs with interesting pharmaceutical properties are currently in clinical trials (*Chand & Karuso, 2017*). Therefore, the study of nudibranch symbiotic bacteria needs to be explored. There is one report of symbiotic bacteria in the vestibular gland, associated with the female reproductive system, and in the egg masses of the nudibranch *Dendrodoris nigra* (*Klussmann-Kolb & Brodie, 1999*). In 2012, rod-shaped Gram-negative bacteria were also found in the epithelial cells of the notum and the mantle edge of *D. nigra* (*Zhukova & Eliseikina, 2012*).

In the present study, we explore the possibility of finding anti-MRSA and antiviral molecules from nudibranch symbiotic bacteria. A total of 20 nudibranch species were collected in Bali and the national park of Karimunjawa, Jepara, in the Java Sea. Symbiont bacteria were isolated from stomach tissue and subsequent, research on biological activities was performed using the disk-diffusion agar plate method with supernatants of lysed bacteria. We then investigated the anti-MRSA and antiviral activity of the isolated symbiont bacteria after culturing it.

## MATERIALS AND METHODS

### Sampling

This research used samples that were collected from Lovina, Bali and the National Park of Karimunjawa (5°47′21.6″S 110°3041.9″E) (March 2017), Jepara Indonesia by SCUBA diving 15–20 m in depth, at 18–25 °C. The samples of nudibranchs were put into sterile plastic bags, filled with seawater and stored in a cool box (*He et al., 2014*; *Sabdono et al., 2015*), and immediately brought to the laboratory to be identified, to isolate and to purify the bacteria.

## Symbiont bacteria isolation and collection

The nudibranchs were washed three times, using sterile seawater, to remove the impurities still that attached to the nudibranchs body. The contents of the stomach called viscera and the body were separated by using a sterile knife and were put in a conical tube that contained sterile seawater. Bacterial isolation was done using the serial dilution method ($10^{-1}$, $10^{-2}$, $10^{-3}$, $10^{-4}$) (*Anand et al., 2006*). The bacterial inoculation was performed on the ZoBell Marine Agar 2216 (Himedia, Mumbai, India) and was incubated at a temperature of 37 °C for 5 days. Each bacterial colony that grown on a plate was separated according to shape, elevation and color. Isolation and purification processes were done in the Tropical Marine Biotechnology Laboratory at Diponegoro University. The pure colonies were maintained in slant cultures at 4 °C and in gliserol at −20 °C.

## Anti-MRSA screening

The antibacterial test was conducted using the overlay method, 245 isolates of symbiont bacteria of nudibranchs were cultured in ZoBell 2216E agar media and incubated at 37 °C for 24 h (*Radjasa et al., 2007b*). The pathogenic bacterium *S. aureus* (Strain MRSA from Kariadi Hospital in Semarang, Indonesia, resistant for oxacillin, gentamicin, benzylpenicillin, ciprofloxacin, levofloxacin, tetracycline) was cultured in Muller–Hinton broth (Difco, Detroit, MI, USA) with a shaker condition of 150 rpm at 37 °C for 24 h. The antibacterial test was conducted by mixing MRSA (concentration adjusted using 0.5 McFarland standard) in soft agar media and pouring into the petri dishes containing the marine bacterial colonies. All screening was performed in triplicate experiments. Clear zones were observed following overnight incubation at 37 °C. The clear zone around bacterial colonies, was an indicator for the occurrence of antibacterial activity (*Defer, Bourgougnon & Fleury, 2009*). Isolates that exhibited anti-MRSA activity were chosen for the characterization of bioactive compounds produced, molecular identification, cytotoxicity and anti-HSV activity.

## Bioactive compound extraction

Active isolates of bacterial symbionts of nudibranchs against MRSA were cultured in 3 l that contained of dissolving the nutrient broth powder (Oxoid, NY, USA) in seawater for 6 days (*Singh et al., 2014*). The bacterial suspension was centrifuged with modification method at 5,000 rpm, at 4 °C for 10 min (*Hayashida-Soiza et al., 2008*). The supernatant was extracted with ethyl acetate. The bacterial pellet was extracted with methanol. The ratio of solvent to culture was 1:1 (v/v) (*Kontiza et al., 2008*). Ethyl acetate and methanol fractions were evaporated, the mass was determined and the crude extracts were kept at −20 °C (*Yoghiapiscessa, Batubara & Wahyudi, 2016*).

## Antibacterial assay

The antibacterial activity was evaluated using the Disc Diffusion method (*Montalvão, Singh & Haque, 2014*), according to the Clinical and Laboratory Standards Institute (*CLSI, 2017*). MRSA was swabed on to Muller–Hinton agar medium (Sigma-Aldrich, St. Louis, MO, USA) with the total concentration of 0.5 Mc Farland standart. The paper

disk (6 mm; Advantec, Japan) containing 15 μL of crude extract was placed on the surface of the agar plate culture. The concentrations of crude extract were 50, 250, 500 and 1,000 μg/ml. Ethyl acetate and methanol were used as negative control. Vancomycin was chosen as positive control (Sigma-Aldrich, St. Louis, MO, USA). The culture plates were incubated overnight at 37 °C. Active isolates were shown by the clear zone around the disk (*Redwan et al., 2016*).

## Polymerase chain reaction amplification

DNA extraction was done by chelex method (*De Lamballerie et al., 1992*). This method is a modification used for marine bacteria (*Lee et al., 2003*). The DNA template was used for Polymerase Chain Reaction (PCR) amplification. The primers selected to amplify the 16S rRNA gene segment were: 27f (5′-AGAGTTT-GATCMTGGCTCAG-3′) and 1492r (5′-TACGGY-TACCTTGTTACGACTT-3′) (*Weisburg et al., 1991*). The PCR mixture contained GoTaq®Green Master Mix Promega (12.5 μl), primer 27 F (one μl), primer 1492 R (one μl), template DNA (one μl) and ddH2O (9.5 μl), so that the total volume was 25 μl. The primer concentration was 10 pmol/μl. PCR condition was done by denaturation at 95 °C for 3 min, annealing at 53.9 °C for 1 min, extension 72 °C for 1 min for 30 cycles. The PCR products were examined using 1% agarose gel electrophoresis and the result was visualized by using UVIDoc HD5 (UVITEC, Cambridge, England).

## DNA sequencing and phylogenetic analysis

DNA sequencing was carried out in the PT. Genetica Science (Jakarta, Indonesia). The gene sequences were analyzed using Basic Local Alignment Search Tool (BLAST) (*Altschul et al., 1997*). To identify different species, phylogenetic trees were constructed using MEGA 7 with the 1,000× bootstrap test. The results of BLAST Homology were deposited to the DNA Data Bank of Japan (DDBJ, www.ddbj.nig.ac.jp) in order to obtain the accession number.

## Cells and viruses

African green monkey kidney cells (Vero, ATCC CCL-l81) were grown in Eagle's minimum essential medium (MEM; Eurobio, Les Ulis, France) supplemented with 8% fetal calf serum (FCS; Eurobio, Les Ulis, France) and 1% of antibiotic PCS (penicillin 10,000 IU/ml, colimycin 25,000 IU/ml, streptomycin 10 mg/ml; Sigma, St. Louis, MO, USA). Cells were routinely passaged every 3 days (*Ben Sassi et al., 2008*).

A virus stock of HSV-1, wild 27 strain ACVˢ PFAˢ was obtained from Pr. Henri Agut, Laboratoire de Virologie de la Pitié Salpêtrière, Paris, France. Virus stock was prepared by incubating Vero monolayers (75 cm$^2$ culture flasks seeded with $3.5 \times 10^5$ cells/ml) at low multiplicity and incubating at 37 °C, in 95% air, 5% $CO_2$ (v,v) atmosphere. Three days after infection, the cultures were frozen and thawed twice, before clearing the preparation by centrifugation at a low speed to remove cell debris. The resulting supernatant aliquot was stored at −70 °C until further used. Virus titrations were performed by the Red and Muench dilution method, using 10 wells on 96-wells microtiter plates per dilution.

 

The virus titre was estimated from cytopathogenicity and expressed as 50% infectious doses per milliliter ($ID_{50}$/ml).

## Cytotoxicity and antiviral test

Using the Vero cell/HSV-1 model, cytotoxicity was evaluated by incubating 100 μl of cellular suspension ($3.5 \times 10^5$ cells/ml) with various dilutions (concentration 5–1,000 μg/ml) of potential extract (NP31-01 and KJB-07) in 96 well plates in Eagle's MEM with a total volume of 200 μl. Vero cells were placed in Eagle's MEM with a final volume of 200 μl in each well and were used as positive controls. The well plates were incubated at 37 °C for 72 h. Cytotoxic activity was observed using microscope and cells were tested using the neutral red dye method. Optical density was measured at 540 nm using a spectrophotometer (SpectraCount$^{TM}$; Packard, Paris, France). The 50% cytotoxic concentration ($CC_{50}$) of the test compound was defined as the concentration that reduced the absorbance of mock-infected cells to 50% of that of controls. $CC_{50}$ values were determined as the percentage of destruction (%D): (($OD_c$)C − $OD_c$)MOCK/($OD_c$)C) × 100. ($OD_c$)C − ($OD_c$)MOCK were the OD values of the untreated cells and treated cells (*Langlois et al., 1986*).

Using the Vero cell/HSV-1 model, 100 μl of cellular suspension ($3.5 \times 10^5$ cells/ml) in Eagles's MEM containing 8% FCS were incubated with 50 μl of a dilution of extracts (concentration 5–1,000 μg/ml) in 96 well-plates (48 h, 37 °C, 5% $CO_2$) Three replicates were infected using 50 μl of medium and a HSV-1 suspension at a multiplicity of infection (MOI) of 0.001 $ID_{50}$/cells. Acyclovir (9-(2-hydroxyethoxymethyl) was used as a control positive against HSV-1 ranging from 0.05–5 μg/ml. Cultures were grown in incubation at 37 °C for 72 h in a humidified $CO_2$ atmosphere (5% $CO_2$). Antiviral activity was observed using Olympus model CX23LEDRFS1 Optical Microcope and cells were tested using the neutral red dye method. The protection of the extract from virus-infected cells was expressed by 50% effective antiviral extract concentration ($EC_{50}$). OD was measured at 540 nm. The OD was related directly to the percentage of viable cells, which was inversely related to the cytopathic effect. $EC_{50}$ values were determined as the percentage of cell protection (%P): (($OD_t$ virus − $OD_c$ virus)/($OD_c$ MOCK − $OD_c$ virus)) × 100. $OD_c$ and $OD_t$ were the OD values of the virus control and test sample, $OD_c$ MOCK was the OD of mock-infected control (*Langlois et al., 1986*). Interpretation of the data was presented using a system of linear regression equations.

## TLC and LC-MS/MS analysis

The compounds were analyzed using Thin Layer Chromatography with silica gel on aluminum sheets (20 × 20 cm) $F_{254}$ KgaA. Gradient: hexane and ethyl acetate (7:3, 1:1, 3:7). Compound bands were detected by UV absorbance and fluorescence. The compounds were characterized using Liquid chromatography–mass spectrometry (LCMS)/MS (*Choma & Jesionek, 2015*). LC-MS/MS analyses were done on a UNIFI chromatographic instrument, with an acquity UPLC$^®$ HSS T3 1.8μ (2.1 × 100 mm) column. Mobile phase A was 0.1% formic acid/water and mobile phase B was acetonitrile + 0.1 formic acid with gradient A/B = 95/5, 60/40, 0/100 and 95/5 in 10 min.

**Table 1 Samples collected and total number of symbiont bacteria in Nudibranch.** The table show the morphology identification of the nudibranch. The bold entries in Table 1 is a nudibranch that has the potential to produce symbiont bacteria that have bio-activity.

| Species code | Nudibranch species | Appearance of the opisthobranch | Number of isolates of bacteria | Depth (m) | Substrate | Locality |
|---|---|---|---|---|---|---|
| NT1-162 | *Hypselodoris whitei* | Whitish background color with reddish purple longitudinal lines covering the mantle. The rhinophores are orange to orange–red with a distinctive white tip. The gills are similarly colored with white on the inside and usually at the tip of each gill. | 10 | 19 | Sand | Bali |
| NT1-164 | *Hypselodoris infucata* | Gray mantle with yellow and purple spots spread on the surface. The rhinophores are red and the gills composed by two-dimensional leaf with a red line along the internal and external edge. | 11 | 15 | Hydroid | Bali |
| NP31-04 | *Goniobranchus leopardus* | The mantle consists of purple–brown marks, ringed with reticulate brownish background, and a purple border. There are four color bands around the mantle edge, an outermost white, then translucent grayish purple, then white, then yellow. | 12 | 18 | Coral | Bali |
| NT31-03 | *Thuridilla gracilis* | Dark background, fine white longitudinal lines and the foot, head and rhinophores edged with bright orange. The parapodia are bordered with bright orange tips. | 8 | 17 | Sand | Bali |
| NP31-01 | **Phyllidia varicosa** | **The mantle consists of longitudinal, tuberculate notal ridges. The ridge and bases of the tubercles are blue–gray in color and the tubercles are capped in yellow.** | **10** | **15** | **Coral** | **Bali** |
| KJN-17 | *Phyllidia ocellata* | The dorsal pattern consists of series of white tubercles, dark background and yellow rims. | 19 | 15 | Hydroid | Karimunjawa |
| KJN-09 | *Phyllidia varicosa* | The mantle consists of longitudinal, tuberculate notal ridges. The ridge and bases of the tubercles are blue–gray in color and the tubercles are capped in yellow. | 19 | 15 | Sand | Karimunjawa |
| KJT-02 | *Caloria indica* | The body is full of white cerata and the horns are white with few purple rings spread along the axes. | 10 | 15 | Sponge | Karimunjawa |
| KJN-13 | *Phyllidiella nigra* | The black background mantle is ornamented by white clustered tubercles. | 12 | 17 | Hydroid | Karimunjawa |
| KJB-07 | **Phyllidia coelestis** | **Black background mantle with blue–gray ridges; yellow-capped mid-dorsal tubercles.** | **18** | **15** | **Coral** | **Karimunjawa** |

*(Continued)*

| Species code | Nudibranch species | Appearance of the opisthobranch | Number of isolates of bacteria | Depth (m) | Substrate | Locality |
|---|---|---|---|---|---|---|
| | | **Table 1** (continued). | | | | |
| KJN-08 | *Phyllidia varicosa* | The mantle consists of longitudinal, tuberculation total ridges. The ridge and bases of the tubercles are blue–gray in color and the tubercles are capped in yellow. | 19 | 15 | Sponge | Karimunjawa |
| KJN-18 | *Phyllidiopsis pipeki* | The rhinophores are pink with a black tip, black lines extending from pale brwon mantle edges. | 15 | 15 | Sponge | Karimunjawa |
| KJN-19 | *Phyllidiella striata* | White clustered tubercles and few yellow caps; black background dorsal. | 8 | 15 | Coral | Karimunjawa |
| KJN-47 | *Phyllidia varicosa* | The mantle consists of longitudinal, tuberculate notal ridges. The ridge and bases of the tubercles are blue–gray in color and the tubercles are capped in yellow. | 12 | 16 | Sponge | Karimunjawa |
| KJN-46 | *Phyllidia varicosa* | The mantle consists of longitudinal, tuberculate notal ridges. The ridge and bases of the tubercles are blue–gray in color and the tubercles are capped in yellow. | 11 | 18 | Tunicata | Karimunjawa |
| KJN-48 | *Phyllidiopsis shireenae* | White dorsal with white tubercles; two longitudinal black lines. | 11 | 14 | Sponge | Karimnujawa |
| KJN-42 | *Goniobranchus kuniei* | They all have large purple or purple–brown spots or marks, usually ringed with white, brownish background, and a purple border. | 12 | 16 | Coral | Karimunjawa |
| KJN-44 | *Phyllidiella cooraburrama* | Extremely large, isolated, notal tubercles and black dorsal background. | 10 | 18 | Coral | Karimunjawa |
| KJN-5 | *Phyllidiopsis shireenae* | White dorsal with white tubercles; two longitudinal black lines. | 8 | 15 | Bryozoa | Karimunjawa |
| KJN-45 | *Phyllidiopsis pipeki* | The rhinophores are pink with a black tip, black lines extending from pale brown mantle edges. | 10 | 15 | Coral | Karimunjawa |

# RESULTS

## Nudibranchs identification

Nudibranchs are widely spread in the seas around the world, especially in Indonesia (http://www.marinespecies.org/imis.php?persid=7211). A total of 15 nudibranchs species were collected from the National Park of Karimunjawa, Jepara and five from Bali. Several nudibranchs have been found on various basic substrates. Most of them feed on sponges, tunicates, hydroids and bryozoans (*Avila, 2006*). The species were identified based on color, shape of the body and surface texture of the body as shown in Table 1 (*Behrens, 2005*; *Coleman, 2001*). The identification results showed that 20 nudibranch species collected from Karimunjawa and Bali seawaters belonged to four genera, three
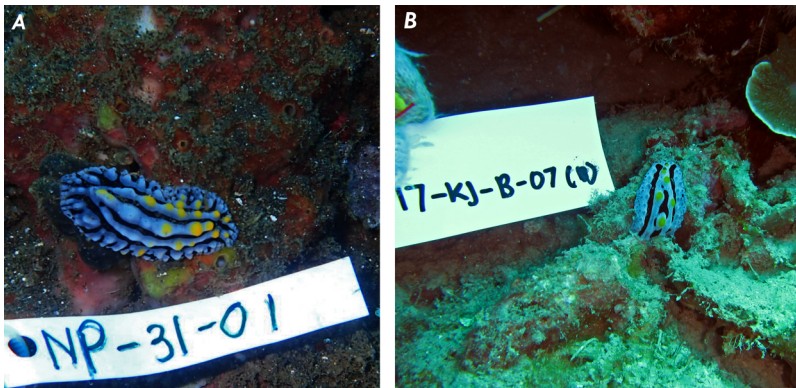

**Figure 1** Underwater photographs of the nudibranch: (A) *Phyllidia varicosa* from Bali; (B) *Phyllidia coelestis* from Karimunjawa, Jepara Indonesia.

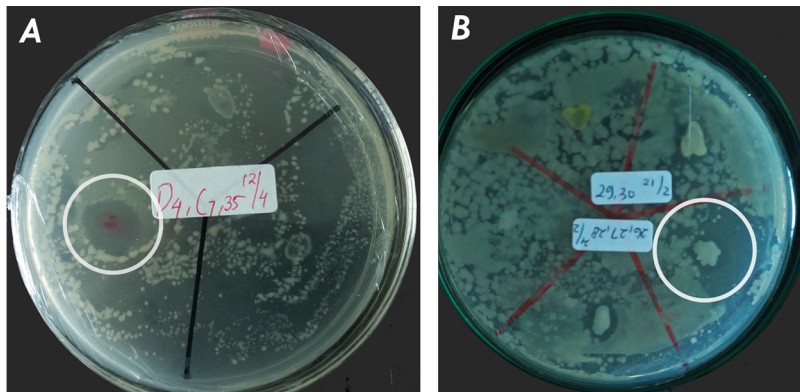

**Figure 2** Active isolate symbiotic bacteria of nudibranchs against MRSA by using agar overlay method: (A) KJB-07 symbiotic *Phyllidia coelestis*; (B) NP31-01 symbiotic *Phyllidia varicosa*.

nudibranch Phyllidiidae, Chromodorididae, Flabellinidae and Plankobranchidae (Sacoglossa) (Table 1). Examples are shown in Fig. 1.

## Isolation of symbiont bacteria

Preliminary isolation of bacterial symbionts from the surface and the viscera of nudibranchs were determined by the growth of bacteria in Zobell 2216E agar media. The numbers of isolates were determined and are presented in Table 1.

## Anti-MRSA screening and crude extract assay

The results of the antibacterial activity are presented in Fig. 2. There is one active symbiont isolate of *Phyllidia coelestis* (KJB-07) from Karimunjawa and one active symbiont isolate of *Phyllidia varicosa* (NP31-01) from Bali based on appearance of the clear zone. The total mass of crude extract from supernatant of KJB-07 and NP31-01 was one g/3 l. The ethyl acetate extract was found to be active against MRSA at concentrations of 500 μg/ml and 1,000 μg/ml as shown in Table 2. The methanol extract from the bacterial pellet was found to be inactive against MRSA.

**Table 2 Anti-MRSA activity (overlay, extract, antibiotic).** Activity screen with the following test anti-MRSA. The signs use indicate: −, no zone of inhibition (ZOI); +, ZOI 7–8.5 mm; ++, ZOI 8.6–10 mm; +++, ZOI 11–15 mm; nt, not tested.

| Crude extract | Concentration (µg/ml) | Diameter of Clear zone (mm) of anti-MRSA | | | | |
|---|---|---|---|---|---|---|
| | | KJB-07 | NP31-01 | KJB-07 | NP31-01 | Vancomycin |
| Bacterial pellet (Methanol) | 50 | − | − | − | − | nt |
| | 250 | − | − | − | − | nt |
| | 500 | − | − | − | − | nt |
| | 1,000 | − | − | − | − | nt |
| Supernatant (Ethyl acetate) | 50 | − | − | − | − | nt |
| | 250 | − | − | − | − | nt |
| | 500 | − | − | + | + | nt |
| | 1,000 | − | − | ++ | + | nt |
| Vancomycin | 30 | nt | nt | nt | nt | +++ |

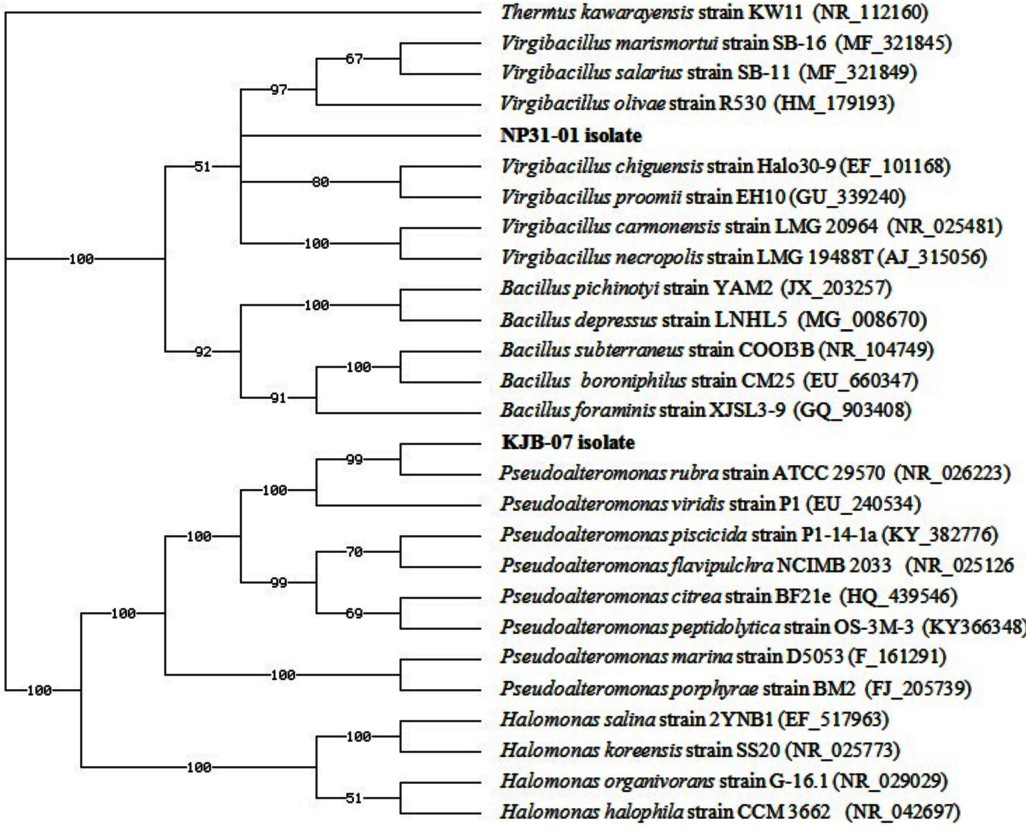

**Figure 3 Phylogenetic tree of bacteria associated with nudibranch isolated from Karimunjawa National Park and Bali, Indonesia.** *Thermus kawarayensis KW11* was used as the out-group.

**Table 3 Evaluation of cytotoxic and anti-HSV-1 activity of ethyl acetate extract of supernatant from *P. rubra* and *V. salarius*.**

| Extract of Bacteria species | $CC_{50}$ (μg/ml) | $EC_{50}$ (μg/ml) |
|---|---|---|
| *Pseudoalteromonas rubra* | 75 | – |
| *Virgibacillus salarius* | >1000 | – |
| Acyclovir | >5 | 0.5 |

**Notes:**
(−), no cytotoxic effect.
$CC_{50}$, the 50% cytotoxic concentration of extract in Vero cells (μg/ml).
$EC_{50}$, concentration of compound in μg/ml producing 50% inhibition of virus-induced cytopathic effect.

## DNA sequencing and phylogenetic tree

The complete 16S rDNA sequences of strain NP31-01 (1,400 bp) and strain KJB-07 (1,376 bp) were obtained. The sequence homology of 99% in phylogenetic tree showed that the strain KJB-07 from *Phyllidia coelestis* was closely related to that of *Pseudoalteromonas rubra* (Fig. 3) and the strain NP31-01 from *Phyllidia varicosa* was closely related to *Virgibacillus salarius* (Fig. 3). This sequence has been deposited to the Genbank. The sequences have received the accession number LC328972.1 (KJB-07) and MH016561 (NP31-01).

## Cytotoxic and antiviral activity

After 3 days of treatment, microscopically visible alteration of normal cell morphology was observed and the viability assay showed destruction of a cell layer for extract from *Pseudoalteromonas rubra*. This extract was cytotoxic on the Vero cell in the entire range of concentration assayed (5–1,000 μg/ml) as shown in Table 3. At concentrations of 1,000 μg/ml, 71% of cell destruction was observed. At MOI 0.001 $ID_{50}$/cells, no anti HSV-1 was performed after 72 h.

The compounds from *V. salarius* present no cytotoxicity in the entire range of concentrations assayed (5–1,000 μg/ml) as shown in Table 3. At concentrations of 1,000 μg/ml only 10% of cell destruction was shown but no anti HSV-1 was present.

## TLC and LC-MS/MS data

The results showed that the three compounds were well separated on Thin layer chromatography (Fig. 4). Of the three compounds produced by *Pseudoalteromonas rubra* in (KJB-07) LC-MS/MS (Fig. 5; Table 4), compound **1**, stearidonic acid showed ions at *m/z* 277.2164; compound **2**, prodigiosin showed ions at *m/z* 324.2900; and compound **3**, (22E,24R)-5α8α-Epidioxyergosta-6,9,22-trien-3β-ol showed ions at *m/z* 449.3118. The chemical structures are given in Fig. 6.

## DISCUSSION

In this paper, we present the first study on anti-MRSA, cytotoxicity and antiviral activities from crude extracts of symbiotic bacteria in Heterobranchia. Nudibranchs, are one of many marine invertebrate groups that feed on marine sponges, a source of bioactive compounds. The bioactive compounds found in nudibranchs are greatly influenced by food and symbiotic bacteria (*Fisch et al., 2017*). Nudibranchs have a wide variety of

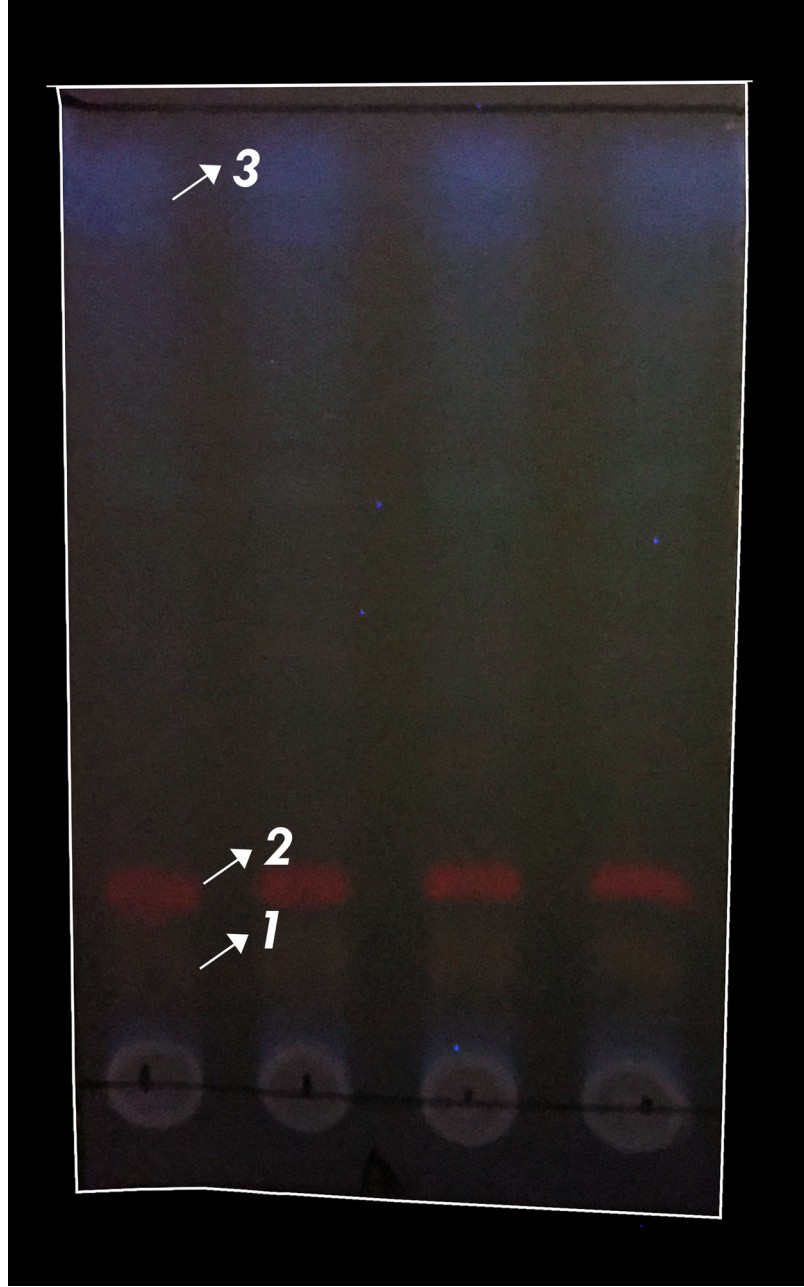

**Figure 4** **Typical TLC photography of ethyl acetate extract of *P. rubra*.** Four repetition of TLC visualization (1) stearidonic acid, (2) Prodigiosin, (3) (22E,24R)-5α8α-Epidioxyergosta-6,9,22-trien-3β-ol.

bacteria that are affected by location and food. In previous studies, different habitats and conditions have been shown to affect the biodiversity of symbiotic bacteria and bioactive compounds (*Zhukova, 2014*). We found different activities of two potential symbiotic bacteria of nudibranchs from the two locations studied. Many species that were found in Karimunjawa are from the same family, which is *Phyllididae*, whereas in Bali a wide range of species were found from different families. Previous work reported that *Phyllididae*

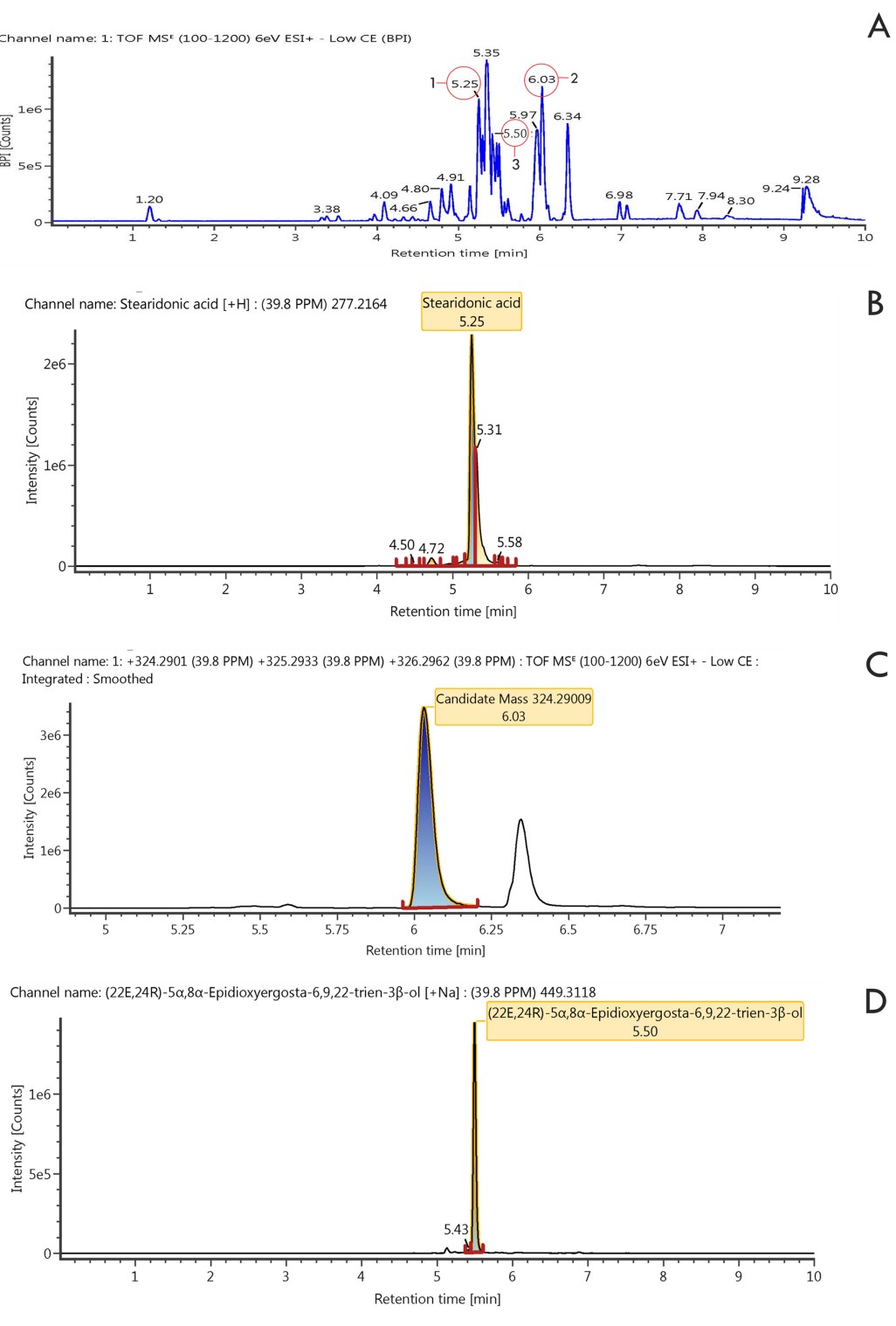

**Figure 5  LSMS/MS Chromatogram of ethyl acetate extract and three compounds.** (A) Extract of ethyl acetate, (B) stearidonic acid, (C) Prodigiosin, (D) (22E,24R)-5α8α-Epidioxyergosta-6,9,22-trien-3β-ol.

**Table 4 Identification of compound in *P. rubra.*** Rt, retention time; The exact mass (calc.) according to the molecular formula of identified compound.

| Compound | Compound name | Observed RT (min) | Molecular formula | Exact mass (Calc.) |
|---|---|---|---|---|
| 1 | Stearidonic acid | 5.25 | $C_{18}H_{28}O_2$ | 277.2164 |
| 2 | Prodigiosin | 6.03 | $C_{20}H_{37}NO_2$ | 324.2900 |
| 3 | (22E,24R)-5α8α-Epidioxyergosta-6,9,22-trien-3β-ol | 5.50 | $C_{28}H_{42}O_3$ | 449.3118 |

a) $C_{18}H_{28}O_2$          b) $C_{20}H_{37}NO_2$          c) $C_{28}H_{24}O_3$

**Figure 6 The chemical compounds.** (A) compound **1**, stearidonic acid; (B) compound **2**, prodigiosin; (C) compound **3**, (22E,24R)-5a8a-Epidioxyergosta-6,9,22-trien-3b-ol.

commonly produce new sesquiterpenoid isonitriles and several other bioactive compounds (*Gastropoda et al., 2018*). We report here that symbiotic bacteria from the family *Phyllidiidae* have potential anti-MRSA and cytotoxicity activity.

Earlier studies were limited by the fact that entire animals were used for isolation of bioactive compound extracts (*Dewi et al., 2016*; *Sim et al., 2018*; *Zhukova, 2014*). In contrast, we cultured the symbiont bacteria that produce the active compounds, allowing us to produce large amounts of extract while conserving the marine environment by using only a very limited number of animals.

We isolated bacteria from the body surface and internal organs (known as viscera) of nudibranchs. We have identified two bacteria species, which show potential anti-MRSA activity.

In previous studies, bacteria have been successfully isolated from the epithelial cells of the notum, the mantle edge and vestibular gland (*Klussmann-Kolb & Brodie, 1999*; *Zhukova & Eliseikina, 2012*). It has been suggested that the bacteria may play a role in protection or defense from predators (*Fajardo et al., 2014*), however, the specific function of these bacteria is still under discussion. No molecular identification of bacteria has been reported to date. In the screening effort of our studies, we have identified symbiotic bacteria as capable of producing anti-MRSA compounds through the complete sequencing of the 16S rRNA gene. Two bacterial isolates were found to have 99% sequence identity with *Pseudoalteromonas rubra* (KJB-07) and *V. salarius* (NP31-01), respectively. Previous work reported that *Pseudoalteromonas rubra*, which had been successfully isolated from a small piece of marine sponge (*Mycale armata*) producing red pigment, had antibacterial activity (*Fehér et al., 2008*). Our study is comparable with previous research,

in which a novel phenolic anti-MRSA compound from *Pseudoalteromonas phenolic* was isolated from sea water (*Isnansetyo & Kamei, 2003*). Another study on anti-MRSA has been explored in other bacterial symbionts of marine organisms such as Tunicata, Porifera and marine algae (*Fedders, Podschun & Leippe, 2010*; *Kamei & Isnansetyo, 2003*; *Hentschel et al., 2001*). No research on anti-MRSA activity of the symbiotic bacteria of nudibranchs has been reported previously.

To evaluate anti-bacterial activity, we showed that the ethyl acetate extract of the supernatant of the bacteria caused an inhibition zone in MRSA cultures as indicated in Table 2, whereas the methanol extract of centrifuged bacteria showed no activity. The ethyl acetate extract showed that anti-MRSA activity of *Pseudoalteromonas rubra* KJB-07 was slightly higher than *V. salarius* NP31-01 at concentrations of 500–1,000 µg/ml. The diameter of an inhibition zone of *V. salarius* NP31-01 was 7.18 mm at a concentration of 500 µg/ml and increased to 8.26 mm at concentration 1,000 µg/ml. The inhibition zone of *Pseudoalteromonas rubra* KJB-07 increased from 8.15–9.23 mm at concentrations of 500–1,000 µg/ml. The anti-MRSA activity of crude extract was comparable with Vancomycin in that the antibiotic had higher inhibition zone in 15 mm at concentrations of 30 µg/ml.

Table 2 shows that the two extracts had different effects on MRSA. In this research, *Pseudoalteromonas*, a Gram-negative bacteria have the capability of producing substances in a single fermentation because some of the substances are bioactive compounds (*Isnansetyo & Kamei, 2009*). *Pseudoalteromonas rubra* was able to prevent the loss of intracellular proteins, and reduced the access of hydrolytic enzymes and some compounds produced by *S. aureus* (*Pinto et al., 2017*). Extract from *Pseudoalteromonas rubra* KJB-07 had stronger activity than from *V. salarius* NP31-01. Recent studies reported that the ethyl acetate extract from *Pseudomonas* sp. was able to produce the compound 1-acetyl-beta-carboline, which is active against MRSA at concentrations of 32–128 µg/ml (*Lee et al., 2013*). *Isnansetyo et al. (2003)* showed bactericidal and bacteriolytic activity of 2,4-diacetylphloroglucinol (DAPG) produced by *Pseudomonas* sp. against MRSA and Vancomycin resistant *Staphylococcus aureus* (VRSA). The mechanisms of resistance affected the activity of DAPG. The activity of compounds produced by marine bacteria should be further explored and evaluated.

The cytotoxicity in the *Pseudoalteromonas rubra* KJB-07 extracts on Vero cells was directly proportional to the increase in the concentration of the extract. This may be due to production of endotoxins. We analyzed by LCMS to identifiy whether the cytotoxic effect of the *Pseudoalteromonas rubra* KJB-07 extract is due to the same compound that is active against the MRSA or not.

The *V. salarius* NP31-01 extracts had no cytotoxic effect at any of the concentrations tested. *Virgibacillus* sp. has been known as non-cytotoxic and to be a potential source of new polysaccharide bioflocculant (*Cosa et al., 2011*). The cell wall of *V. salarius* contained meso-diaminopimelic acid as a major component which has been suggested to correspond to the non-cytotoxicity active compounds against MRSA and *Hua et al. (2008)*. Previous work reported the mechanisms of bacteria associated with marine organisms, which are bacteria associated with the ascidian *Cystodytes dellechiajei* (*Martínez-García*

*et al., 2007*) and with marine flatworms (*Lin et al., 2017*). Martínez-García et al., reported the role of the bacterial community associated with *C. dellechiajei* in the production of Pyridoacidine alkaloids. This research evaluated that antiproliferative activities are found in the host and the associated bacteria but the cytotoxic activities are only found in the associated bacteria. The research indicated that the associated bacteria transformed the compounds and that it could be toxic due to environmental influences. The crude extract from the bacteria associated with marine flatworms had potential anti-MRSA and toxic activity against Hela-cells. Further study is necessary to separate and identify the active and cytotoxic compound.

Recent studies have demonstrated the importance of antiviral compounds from marine invertebrate; such as *Phyllocaulis boraceiensis*, which contains polyunsaturated fatty acids that disturb the virus envelope (*Toledo-Piza et al., 2016*). Manzamine A, from the sponge genus *Acanthostrongylophora*, works by inhibiting viral replication (*Palem et al., 2011*). In our study, crude extracts of *Pseudoalteromonas rubra* and *V. salarius* showed no activity against HSV-1. After processing and screening several anti-MRSA activities, we obtained results that the extract from *Pseudoalteromonas rubra* KJB-07 constantly showed the antibacterial inhibition. In contrast, extract from *V. salarius* NP31-01 had lost any antibacterial activity. We focused on identifying the compound of crude extract from *Pseudoalteromonas rubra* KJB-07 for further study. We detected several minor compounds of *Pseudoalteromonas rubra* KJB-07 extracts with LCMS and the minor compounds showed the same signal as the control group. Three major compounds were chosen based on different signals compare with the control group. We isolated and identified three compounds from *Pseudoalteromonas rubra* KJB-07, that showed anti MRSA activity and also cytotoxic activity (Table 4) based on the similarity of molecular weight, molecular formula and spectrum patterns compared to those in the LCMS data base (library). These compounds, prodigiosin, stearidonic acid and (22E,24R)-5α8α-Epidioxyergosta-6,9,22-trien-3β-ol, are not novel. Prodigiosin has been shown to have antibacterial and cytotoxic activity (*Francisco et al., 2007*). Another study has shown that stearidonic acid had synergism with amphotericin-B in inhibiting Candida (*Thibane et al., 2012*). The compound (22E,24R)-5α8α-Epidioxyergosta-6,9,22-trien-3β-ol has been found previously in *Hypsizigus marmoreus* (*Xu et al., 2007*) and in the South-East Asian mushroom *Amanita subjunquillea* (*Kim et al., 2008*). This compound has been shown to have moderate toxicity against human tumor cells.

## CONCLUSIONS

This is the first reported study describing research on the anti-MRSA, cytotoxicity and anti HSV-1 activity of bacterial symbionts from the viscera of nudibranchs. We showed that crude extracts from culture supernatants of *Pseudoalteromonas rubra* and *V. salarius*, symbionts of these nudibranchs, have anti-MRSA activity, which must be caused by compounds that are produced and secreted by the bacteria. Extracts from *Pseudoalteromonas rubra* showed cytotoxic effects on Vero cells, whereas extracts from

*V. salarius* did not show cytotoxic effects. No anti HSV-1 activity was detected in any of the bacterial extracts. The crude extract of *Pseudoalteromonas rubra* had stable anti-MRSA activity compared to the extract from *V. salarius*. Three compounds from *Pseudoalteromonas rubra* were successfully identified: prodigiosin, stearidonic acid and (22E,24R)-5α8α-Epidioxyergosta-6,9,22-trien-3β-ol. Prodigiosin and stearidonic acid have been suggested to exhibit antibacterial activity; moreover epidioxyergosta has been suggested to have cytotoxic activity.

## ACKNOWLEDGEMENTS

The authors thank Kadek Fendi Wirawan for help with collecting samples in Bali and thank to Research Center for Chemistry, Indonesian Institute of Sciences (LIPI), PUSPITEK, Indonesia for providing the LCMS/MS instrument.

### Funding

This work was approved by the Diponegoro University and supported by grants from the Directorate Research and Community Services Ministry of Research Technology and Higher Education Jakarta, Indonesia, the PMDSU (Program Magister Doktor Sarjana Unggul) (No. 315-12/UN7.5.1/PP/2017) and mobility grant under sandwich-like program (1930/D3.2/PG/2017). This project was also partially financially supported by a PEER grant (Subaward Number: 2000007644) to Ocky Karna Radjasa. The funders had no role in study design, data collection and analysis, decision to publish, or preparation of the manuscript.

### Grant Disclosures

The following grant information was disclosed by the authors:
Diponegoro University.
Directorate Research and Community Services Ministry of Research Technology and Higher Education Jakarta, Indonesia, the PMDSU (Program Magister Doktor Sarjana Unggul): 315-12/UN7.5.1/PP/2017.
Sandwich-like program: 1930/D3.2/PG/2017.
PEER grant (Subaward Number): 2000007644.

### Competing Interests

The authors declare that they have no competing interests.

### Author Contributions

- Rhesi Kristiana conceived and designed the experiments, performed the experiments, prepared figures and/or tables, approved the final draft.
- Gilles Bedoux conceived and designed the experiments, analyzed the data, contributed reagents/materials/analysis tools, prepared figures and/or tables, authored or reviewed drafts of the paper, approved the final draft.

- Gerard Pals analyzed the data, prepared figures and/or tables, authored or reviewed drafts of the paper, approved the final draft.
- I. Wayan Mudianta performed the experiments, authored or reviewed drafts of the paper, approved the final draft, identified nudibranch.
- Laure Taupin performed the experiments, authored or reviewed drafts of the paper, approved the final draft.
- Christel Marty performed the experiments, prepared figures and/or tables, approved the final draft.
- Meezan Ardhanu Asagabaldan performed the experiments, prepared figures and/or tables, approved the final draft.
- Diah Ayuningrum performed the experiments, prepared figures and/or tables, approved the final draft, helped with experiments.
- Agus Trianto conceived and designed the experiments, performed the experiments, authored or reviewed drafts of the paper, approved the final draft.
- Nathalie Bourgougnon conceived and designed the experiments, analyzed the data, contributed reagents/materials/analysis tools, prepared figures and/or tables, authored or reviewed drafts of the paper, approved the final draft.
- Ocky Karna Radjasa conceived and designed the experiments, authored or reviewed drafts of the paper, approved the final draft.
- Agus Sabdono conceived and designed the experiments, authored or reviewed drafts of the paper, approved the final draft.
- Muhammad Hanafi analyzed the data, contributed reagents/materials/analysis tools, authored or reviewed drafts of the paper, approved the final draft.

## Field Study Permissions

The following information was supplied relating to field study approvals (i.e., approving body and any reference numbers):

Field experiments were approved by Government of Bali Province (070/00178/DPMPTSP-B/2018; 070/01369/BPMP/2016) and the Ministry of Environment and Forestry, Directorate General of the Conservation of Natural Resources and the Ecosystem of Karimunjawa National Park (1048/T-34/TU/SIMAKSI/03/2017).

## DNA Deposition

The following information was supplied regarding the deposition of DNA sequences:

The sequences are available at NCBI: LC328972.1 and MH016561.

## Data Availability

The raw data is available in the tables, figures and Supplemental Files.

## Supplemental Information

Supplemental information for this article can be found online at http://dx.doi.org/10.7717/peerj.8093#supplemental-information.

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
