# Peer review of "Bioactivity of compounds secreted by symbiont bacteria of Nudibranchs from Indonesia"

_PeerJ, doi:10.7717/peerj.8093_

## Round 0.1 · original submission · Major Revisions

· Academic Editor

Major Revisions

The manuscript contains several problems that must be corrected before I can send it to the reviewers:
- figure numbering is wrong throughout: for example, the text refers to Figure 5 while the data show are present in Figure 9 instead.
- The TLC figure appears to be an analysis of 4 identical samples. It should also contain labels showing the identities of lanes/spots.
- The legend to Table 2 does not state the meaning of "-" or "+"
- The LSMS/MS figure seems to be an analysis of pure compounds, whereas an analysis of your mixture is missing.
- How did you ascertain that three compounds isolated are indeed what you claim (prodigiosin, etc.) ?

---

## Round 0.2 · Major Revisions

· Academic Editor

Major Revisions

Both reviewers require stronger data, and I agree with their assessment. At least, you should include (either in the main text or as Supporting Information) pictures of the Disc diffusion assay of the ethyl acetate and methanol extracts.

I still consider that your analysis of the three isolated compounds is confusing. For example, the LSMS chromatogram includes many compounds but you only discuss three. Why? Was their identification impossible? if so, please state that explicitly. It is also surprising that none of those other compounds can be detected in the TLC, even though some of them are more abundant than any of the other three compounds. Please clarify

Reviewer 1 ·

Basic reporting

The title is suitable and reflects the research that has been carried out. The results were fairly discussed with sufficient support from previous findings, and with a good flow. However there are certain parts that need correction and further clarification (as commented in the draft). Moreover, the usage of English language is bad and critically needs improvement. Author should re-write many sentences in the drafts (as highlighted) due to poor English command. Minor spelling mistakes spotted in the draft as highlighted in the draft. The research approaches in this study need to be justified.

Experimental design

The article fits the aims and scope of the journal. Methodology is not clearly described, hence not repeatable by practitioners, especially the bioactive compound extraction method. Some of the method have no citation.

Validity of the findings

The amount of data is not sufficient especially in the antimicrobial analysis part. Agar overlay and disc diffusion method is qualitative analysis. Quantitative analysis of antimicrobial study is needed to support the discussion. The validity of the results is doubtful because, the author used incorrect solvent to dissolve the extract. Also the growth of indicator strains in agar plate during the agar overlay assay is very weak. Agar overlay analysis results is reported in duplicate. No statistical analysis shown.

Additional comments

The English language should be dramatically improved to make sure that the readers can understand the text easily. Certain terms and phrasings are confusing. Some of the confusing sentences need to re-phrased, as highlighted in the draft.

Need to correct the symbol of temperature throughout the draft-correct is: °C

Figure 3
-Need to re-write the caption. Need to state the picture was taken from the agar overlay method or disc diffusion method.
-From the picture, it can be seen that the growth of indicator strains is very week especially in picture A. This indicates some error in the experiment, hence the results are doubtful. In actual scenario, the indicator strains must grow well in the area that are not exposed to test strains. Author should clarify on this. Example of typical inhibition zone picture can be found in the following article doi:10.3390/biomedicines5020031

Figure 2 is not clear
The labelling of (A) and (B) on the photo is missing. Also, there is no flowchart of isolation as mentioned in the caption of Figure 2

Citation format error occurred in many places, especially when author cited more than 1 reference for 1 sentence. The correct way to cite more than 1 reference is for example (citation 1; citation 2;citation 3)

Line 325 to 326 :“The ethyl acetate extract showed that anti-MRSA activity of P. rubra was higher than V. salarius at concentrations of 500-1000 µg/ml.”
The difference is not much. Moreover, in Table 2, both isolates scores “++”. So it is not proper to state that anti-MRSA acivity of P. rubra is higher than V. salaries. Unless, quantitative analysis was done to further confirm it.

After the isolates was identified as P. rubra, the correct way to state the scientific name for the author’s strain is by mentioning the code of the isolate as well e.g. P. rubra KJB-07

What is the justification of screening for the three bioactivity (anti-MRSA, cytotoxicity, and anti-viral) and not other bioactivity like antioxidant, etc). From the results obtained, what will be the potential application of the extract obtained from P. rubra KJB-07?

Line 332 to 336: Table 2 shows that the two extracts had different effects on the MRSA. In this research, P. rubra as a gram-negative bacteria had cell walls that were thinner than V. salarius as gram-positive. Pseudoalteromonas rubra was able to prevent the loss of intracellular proteins, and reduced access of hydrolytic enzymes and some compounds produced by S. aureus

This fact is not correct. Nothing to do with the cell wall of P. rubra. The compounds produced from P. rubra will effect the cell wall of MRSA. This sentence critically needs correction.

Annotated reviews are not available for download in order to protect the identity of reviewers who chose to remain anonymous.

Reviewer 2 ·

Basic reporting

Reviewed article is interesting. It is written in proper English. Structure of article is classic.

Experimental design

Authors studied anti-MRSA, anti-HSV-1 and cytotoxic activity of two extracts obtained from Pseudoalteromonas rubra and Virgibacillus salarius. Both extracts acted anti-MRSA. The extract of P. rubra was cytotoxic to Vero cells, but the extract of V. salarius presented no cytotoxicity. No anti-HSV-1 was observed. Authors isolated also three compounds from P. rubra culture supernatant identified by LC/MS. I'm surprised why Authors not studied activity of these 3 substances? Which substance is responsible for the observed effects? Where is LC/MS for V. salarius?

Validity of the findings

Studies are incomplete. It is lack of LC/MS for V. salarius and were not done studies of activity of compounds obtained from supernatant.

---

## Round 0.3 · Major Revisions

· Academic Editor

Major Revisions

I am mostly satisfied with the scientific changes introduced, but there are numerous language deficiencies thoughout (for example , the second sentence in the introduction is virtually unreadabel "Strengthen in previous research, infectious disease is a current alarming issue, disease outbreak, a measure that includes both sickness and death, have become more frequent, with more varied causes " ). There are also wrong references: By and Numbers, 2018 is really Christensen, J. (2018) "Infections by the numbers", Scientific American 318, 5, 48-49. The whole text should be reviewed by a professionally-proficient English speaker. Further, the last image has a very unclear depiction of the structure of (22E,24R)-5α8α-Epidioxyergosta-6,9,22-trien-3β-ol . Please provide a corrected version.

Reviewer 1 ·

Basic reporting

Sufficient field background

Experimental design

fit the aims of the journal.

Validity of the findings

good

---

## Round 0.4 · accepted · Accept

· Academic Editor

Accept

Thank you for providing the requested changes.